# Reproductive biology of long-whiskered catfish (*Mystus aor*) relative to temperature and precipitation from the Kaptai Lake of Bangladesh

Shifat Ara Noor[1], Muhammad[1], Md. Main Uddin Mamun[1], Fatema Akhter[1,2], Mst. Halima Khatun[3], Md. Mahiuddin Zahangir[1], Md. Moudud Islam[1,4] *

1 Department of Fish Biology and Biotechnology, Faculty of Fisheries, Chattogram Veterinary and Animal Sciences University, Chattogram, Bangladesh, 2 Australian National University, Canberra, Australia, 3 College of Fisheries, Ocean University of China, Qingdao, Shandong, PR China, 4 Department of Zoology, University of Otago, Dunedin, New Zealand

* moududfbb@cvasu.ac.bd

**Data Availability Statement:** All relevant data are within the manuscript and the obtained data is not

## Abstract

Life-history information is necessary for effective conservation, aquaculture development and broodstock management of fishing populations, especially for fish with significant economic exploitation. Aiming to characterise the life-history traits in relation to the environmental parameters of the long-whiskered catfish, *Mystus aor*, a commercially important species from Kaptai Lake, Bangladesh, different life-history data were analysed. The monthly condition factor (K) ranged from 1.15 to 1.41, and the relative condition factor (Kn) ranged from 0.14 to 0.25. The estimated relative fecundity was 18474−55504 eggs/per fish, and the maximal oocyte diameter was found in May (0.83 ± 0.11μm). The lengths at first sexual maturity ($L_m$) for male and female *M. aor* were 49.5 cm and 54.1 cm, respectively. The GSI was peaked in May for both males and females, and histological analysis also revealed the presence of yolk granules and several mature spermatids in April and May in both sexes. Analysis of environmental data over 40 years in the Kaptai lake ecosystem indicate the significant increase in the precipitation rate in spawning month which is highly correlated with the spawning season. The findings of the current study will serve as a baseline for the conservation and broodstock management in the Kaptai Lake of Bangladesh.

## Introduction

Fisheries management and conservation aim to sustainably maintain declining fish populations, addressing environmental issues and overfishing [1]. Understanding species life cycle patterns, habitat, and exploitation status is crucial for effective management, ensuring maximum benefits for fishers and communities [2]. Therefore, knowing the fish and fisheries resources, including the life-history characteristics of commercially important species, is crucial for undertaking any management actions.

**Funding:** This study was funded by a grant from the CVASU-University Grant Commission (grant number: UGC-CVASU_FY-2020-21 to Md M. I.). The funder had no role in study design, data collection and analysis, decision to publish, or preparation of the manuscript.

**Competing interests:** The authors have declared that no competing interests exist.

Kaptai Lake, Bangladesh's largest artificial freshwater lake [3], is a significant source of natural fisheries resources. With a surface area of 68,800 hectares [4], it offers potential for fisheries development and increased annual fish production. The annual fish production of Kaptai Lake was 12,696 metric tonnes in the 2019–2020 fiscal year [5], though the total production was much lower (7,247 metric tonnes) 20 years ago [6]. However, the abundance of large fish species like carp and catfish has dropped, while small forager fish have increased [7]. According to recent data, the abundance of carp is below 1%, with other forager fishes dominating more than 95% of the total production [5, 8]. Overexploitation, invasive species, and water diversion contribute to the unbalance in the lake's natural stocks [9]. The government has enforced a 90-day (1May to31 July) fishing ban to increase fish reserves but fishing still goes on in remote areas due to scarcity of management. However, further study must be done to assure the richness and sustainability of the brood stocks, as well as the appropriate management of this lake.

Fisheries research, stock assessment, artificial breeding, conservation, and management all benefit from an understanding of fish reproductive biology, which includes the timing and duration of spawning, sex ratio, maturation stages, length and age at maturity, oocyte diameter and fertility [10]. Fish reproductive biology is essential for managing fisheries [11], particularly in developing nations like Bangladesh, where managers rely on size at first maturity and the commencement and duration of the spawning season to manage fisheries stocks [12]. Most fish have one or two spawning seasons per year, and the spawning season is timed [13]. The spawning season must coincide with the best environmental conditions for larval survival for a successful reproductive event [14]. Changes in the environmental parameters such as rainfall and temperature interdict the reproductive seasonality in fish [15]. Moreover, fish stock management highly depends on understanding a species' fecundity [16], a term for a fish that is first spawning and an assessment of the capability of stock for reproduction. Therefore, obtaining fundamental biological information about spawning season and fecundity in relation to the environmental data is essential for maintaining an effective conservation and management system.

A small population of any sexually mature animal kept as a source of population replacement or for the creation of new populations is known as broodstock. Natural recruitment and reproduction may restore fish stocks, which are renewable natural resources [17]. Any species' ability to reproduce successfully in challenging environmental conditions depends on its ability to recruit. The availability of sufficient brood fish in nature is also critical to the success of reproductive activities. For that, effective management of the broodstock is necessary to assure the availability of ripe broods throughout the breeding season [17]. For maximal survival, improved gonadal development, and increased fertility, broodstock modification entails modifying the environment around the broodstock [18]. Therefore, studies into the factors that shape and control the population biology of fish species that are commercially important should be conducted, with a particular emphasis on the breeding season.

Catfish are considered an important group of fish because they serve many different roles, including as food fish in aquaculture, as research animals, as ornamentals and for sport fishing [19]. The total production of catfish in Bangladesh was 69,389 metric tonnes in the year 2020 −21, whereas the production of catfish in Kaptai Lake was 217 metric tonnes [5]. Long-whiskered catfish, *Mystus aor*, is one of the naturally occurring and commercially important freshwater catfishes of Bangladesh. The fish, locally known as 'ayre' or 'guziayre', belongs to the order Siluriformes and the family Bagridae [20]. Due to its great nutritional value and high protein content, low number of intramuscular bones [21, 22], high market price and widespread popularity [21, 23], it has been regarded as one of the most loved culinary fishes. Though it prefers riverine habitats, it can also be found in ponds, lakes, tanks, channels, and

reservoirs. Adults and juveniles reside on the river's bottom and marginal areas, fry live in shallow marginal pits connected to the river by channels, and larvae live in nests constructed amid rocks or soft, muddy beds of streams, rivers and large tanks [21]. Kaptai Lake is one of the most important natural habitats of *M. aor*, although the stock is gradually declining due to its geographic position and bottom surface.

Several studies have been conducted on various aspects of *M. aor* morphology [21, 22, 24], feeding [25–28] and breeding biology [26, 29, 30]. Study on the biology of this catfish from Kaptai Lake has not yet been conducted. In the last century, oocyte diameter data [26] was measured for determining the reproductive season of the fish but it is not precise enough for the conservation and management plan as well as broodstock management in today's aspect due to the climatic changes occurred in past years. To improve natural broodstock by identifying their breeding season, more intense observation of the reproductive biology is required. Moreover, there was no comprehensive report on reproductive biology to understand the reproductive seasonality with respect to climatic events of this species. In the present investigation, strategic dimensions such as length-weight relationship, condition factor (K), relative condition factor (Kn), size at first maturity (*Lm*), gonadosomatic index (GSI), hepatosomatic index (HSI), fecundity, oocyte diameter and cyclic observation of the gonadal maturation stages are analysed, so that the specific breeding season and strategy can be understood. Therefore, the overall objectives of this study are to understand the basic biological indices and spawning season of *M. aor* and planning a conservation and management strategy with respect to environmental data in the Kaptai lake ecosystem.

## Material and methods

### Sampling site and collection of samples

Fish samples (both male and female *M.aor*) were obtained from the local Banarupa Bazar fish market and fish landing centre of the Bangladesh fisheries development corporation (BFDC), Rangamati, Bangladesh. During the study period, monthly 10 –17 fish (average weight 1040.2 ± 233.7 gm, and total length 55.09 ± 4.43 cm) of both sexes were collected from November 2020 to October 2021. The collected fish were kept in an insulated iced box and then brought to the Faculty of Fisheries, Chattogram Veterinary and Animal Sciences University (CVASU), for further analysis. All the collected fish samples were taken into consideration for further assessment of reproductive biology and seasonal variations in the maturation pattern in the Kaptai lake.

### Collection of length-weight data and determination of the length-weight relationship

The total length (TL, from the tip of the snout to the end of the caudal fin) and standard length (SL, from the tip of the snout to the caudal peduncle) of each fish was measured by using a measuring scale and recorded in centimetres (cm). The weight of the fish was measured using an electric balance (Redwag, WPT1211NV, Poland) and recorded in grams (g). The length-weight relationship was determined by fitting the data to a potential relationship based on the following exponential equation by Le cren [31].

$$TW = aTL^b$$

Where, TW is the total weight (expressed in g); TL is the total length (expressed in cm); "a" is a coefficient related to body form and "b" is an exponent. After that the measured values of

constant 'a' and 'b' was estimated from the log transformed length and weight for Log TW = log a + b Log TL [31].

## Determination of condition factor (K) and relative condition factor (Kn)

The condition factor was calculated by using the following equation of Lima-Junior [32] as K = W × 100/ $L^3$ (W and L are weight and length respectively). The Kn was determined as Kn = W / ^w (W and ^w are observed and expected weight of fish).

## Determination of gonadosomatic index (GSI) and hepato-somatic index (HSI)

The monthly GSI was calculated by using the equation followed by Kumar et al, 2014 [33] as GSI = (weight of gonad/ weight of fish) × 100 and HSI = (weight of liver / weight of fish) × 100 [34].

## Determination of length at first sexual maturity ($L_m$)

The Lm was estimated by using multiple functions such as the relationship of (i) gonadosomatic index (GSI) vs total length (TL), (ii) modified gonadosomatic index (MGSI) [35] vs TL, and (iii) Dobriyal index (DI) [36] vs. TL (MGSI (%) = (GW/BW − GW) × 100 and DI = $\sqrt[3]{GW}$) (Where, GW and BW is gonad and body weight respectively).

## Determination of fecundity and oocyte diameter

The gravimetric method was used to estimate the fecundity. For this, 25 fishes weighing from 1000−1500g with gravid ovary were used. At first, 3–5 subsamples from different parts of the ovary were cut and then weighed the subsamples. After weighing, the number of eggs was counted for each sample, and fecundity was determined by the following formula:

Fecundity = (Total ovary weight)/(Weight of sub-sample) × No. of eggs in the subsample

The diameter of the oocyte from three samples of each female fish was measured by an ocular microscope (Carl ZEISS Microscope Gmblt, Optica B-190 Series) with installed software (Optika Vision Lite 2.1, Italy).

## Histological analysis of gonads

After dissection of the collected fish, the gonads (both male and female) were collected and preserved in Bouin's fixative for 24h to maintain the integrity of the sample. Gonad samples were collected for each month to histologically observe the cyclic gonadal developmental stages of *M. aor*. On the next day, gonad samples were then transferred to 70% ethanol and kept at room temperature until histological study. When starting the samples for histological assay, the samples were dehydrated by different grades of alcohol (70, 80, 90, 95, 100 and 100%). After dehydration, samples were embedded in paraffin wax. Then, the embedded wax was sectioned using a microtome machine (KD 2258, China). The sections were stained by a standard staining procedure using hematoxylin and eosin and then subjected to a histological examination under a microscope (Carl ZEISS Microscope Gmblt, Optica B-190 Series, USA).

## Collection of environmental data

Monthly environmental data, including temperatures, humidity/precipitation, solar fluxes, parameters for solar cooking, and wind/pressure were obtained from the power data access viewer [https://power.larc.nasa.gov/data-access-viewer/] from NASA for the Kaptai Lake

ecosystem (latitude:22.5928, longitude:92.2138) from the year 1980 to 2020. The collected data were correlated with the GSI values and the gonadal maturation pattern of *M. aor*.

## Statistical analysis

Values are presented as means ± standard deviation (SD). The Shapiro-Wilk test of normality was performed to observed the homogeneity and distribution of monthly collected data. Pearson regression ($r^2$) analysis was conducted to establish the correlation between the GSI and HSI. Statistical analysis was performed using Microcal Origin, SPSS, Microsoft Excel, and data were visualised using Microsoft PowerPoint. The GSI and oocyte data gathered in various months were compared for significant differences using one-way analysis of variance (ANOVA). Student t-tests were used to determine whether there were any significant differences in K, Kn and HSI between male and female *M. aor* and also to compare the environmental data between two period of time (1981–1990 and 2012–2021).

## Results

### Length-weight relationship

The logarithmic equation using total body weight (TW) and standard length (SL) was found as TW = 3.29SL– 2.36 ($r^2$ = 0.82) for male *M. aor* collected from Kaptai Lake, Bangladesh. The slope (b) and intercept (a) for males were 3.29 and 2.36, respectively. For females, the logarithmic equation was TW = 2.45 SL– 0.99 ($r^2$ = 0.74). The slope (b) and intercept (a) for females were 2.45 and 0.99, respectively (Fig 1A and 1B). The established logarithmic equation using pooled data (male and female) was TW = 2.65SL– 1.31 ($r^2$ = 0.82) while the slope (b) and intercept for the pooled data were 2.64 and −1.31, respectively (Fig 1C).

### Condition factor and relative condition factor

The monthly K values of collected *M. aor* from Kaptai Lake varied between 1.15 and 1.41, and the Kn value varied between 0.14 and 0.25. The K was peak in May (1.41 ± 0.57) and lowermost in November (1.15 ± 0.05), whereas the Kn value was maximum in July (1.07 ± 0.11) and minimum in November (0.86 ± 0.16) (Fig 2A). Significant differences ($p < 0.01$) were found in the K values between male and females in the month of May (Fig 2A).

For males, the K value peaked in March (1.50 ± 0.08), and for females, it peaked in November (1.52 ± 0.69). The lowest K value was found in November (0.91 ± 0.02) for males; however, it was lowest in July (1.14 ± 0.07) for females. On the other hand, the maximum and minimum Kn values were in April (1.14±0.06) and March (0.86 ± 0.02), respectively, for males. For females, the maximum Kn was in July (1.13 ± 0.4), and the minimum was in November (0.85 ±0.07) (Fig 3). The student t-test found significant differences ($p < 0.05$) in Kn values between male and female only in May (Fig 2B).

### Hepato-somatic index (HSI)

All through the year, monthly variations in the HSI values for *M. aor* were recorded. For pooled data, the HSI value ranged from 0.60 to 1.03, with maximum value in May (1.03 ± 0.26) and the minimum in November (0.60 ± 0.12). For males, the value of HSI ranged from 0.30 to 1.13, with the highest HSI value in March (1.16 ± 0.07) and the lowest in February (0.52 ± 0.15). In females, the value of HSI ranged from 0.52 to 1.16, with the uppermost value (1.13 ± 0.25) in May and the lowest (0.67 ± 0.27) in October (Fig 3). Significant difference was observed between male and females in the month of February ($p < 0.01$), and April ($p < 0.05$).

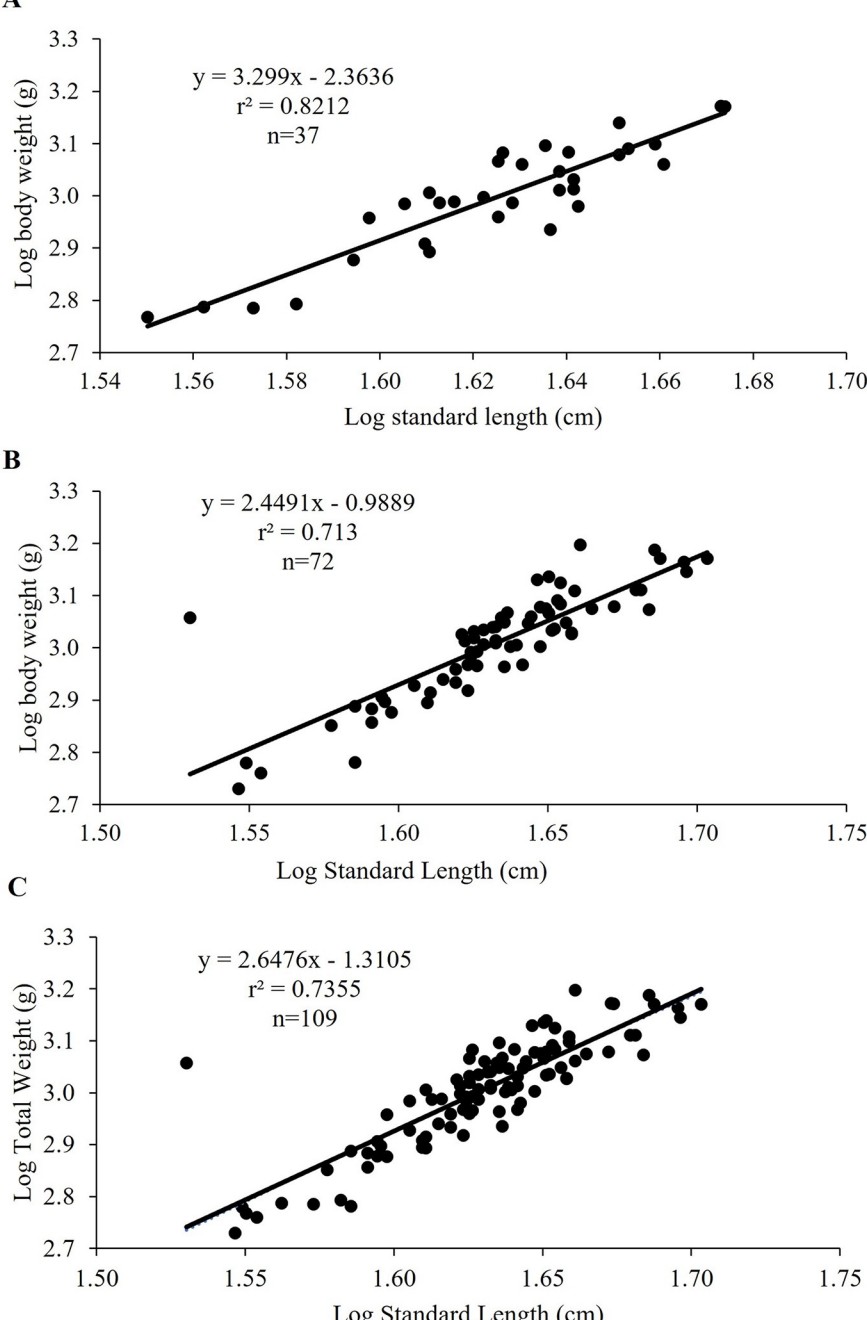

**Fig 1.** Logarithmic relationship between the standard length and weight of (A) male, (B) female and (C) pooled data of *Mystus aor* collected from Kaptai Lake, Bangladesh; X-axis indicates the log-transformed standard length (SL), Y-axis indicates the log-transformed total weight (g) of the fish, and $r^2$ indicates the coefficient of determination (n = 37 for males, n = 72 for females and n = 109 for pooled data).

### Gonadosomatic index (GSI) and oocyte diameter

Monthly changes in the GSI values were observed for *M. aor* throughout the year for both sexes and found significant differences between months ($p < 0.05$). The value of GSI ranged from 0.03–0.78 in males and 0.23–2.32 in females. The highest value of GSI was recorded at

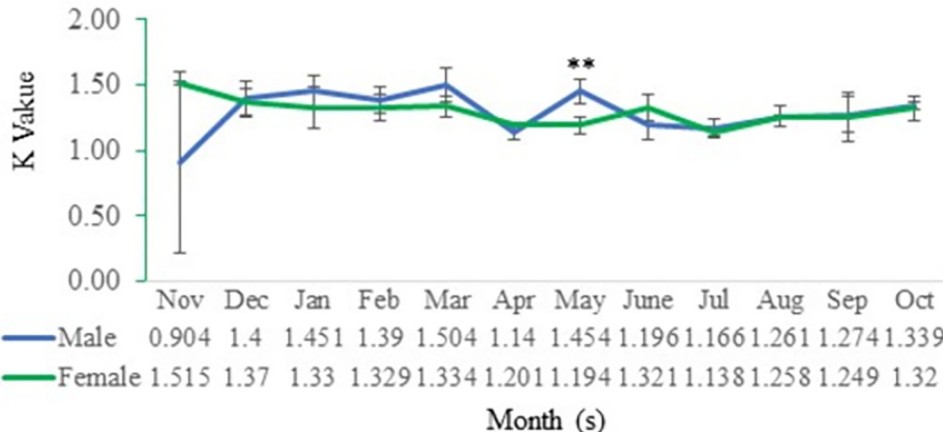

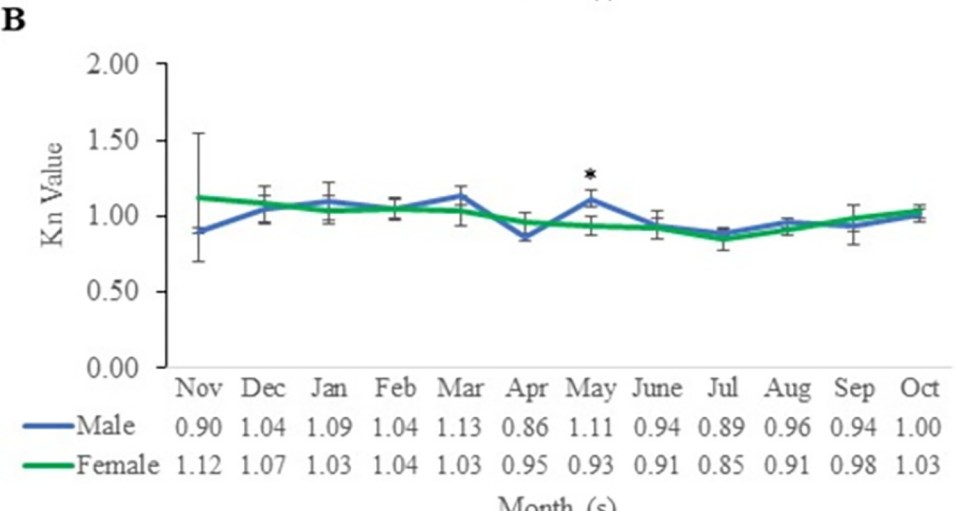

**Fig 2.** (A) Condition factor and (B) relative condition factor of male and female *M. aor* collected from Kaptai Lake, Bangladesh; blue colour indicates males and the green colour indicates females; data are presented as mean ± standard deviation (SD) (n = 5−9 for each sex in each month). Asterisks denotes the significant difference between male and female (*, p < 0.05; and **, p < 0.01).

0.91 ± 0.05 in May, and the lowest value was 0.02 ± 0.002 in October in males (Fig 4C). In females, the peak GSI was found in May (2.32 ± 0.37), and the lowest was in November (0.23 ± 0.21) (Fig 4A).

Oocyte diameter was measured from the monthly collected female *M. aor* using a light microscope and also found statistically different between months (p < 0.05). The highest oocyte diameter was recorded in May (0.83 ± 0.11 μm, Fig 4B). There was a strong positive correlation ($r^2$ = 0.90, p 0.283 and F = 1.456) between the monthly oocyte diameter and the GSI of *M. aor* (Fig 4D).

## Fecundity

A random sample of gravid female fish weighing between 536.8 and 1480.4g, with total lengths ranging from 46.7 to 59.2cm and ovaries weighing between 4.21 and 41.21g, was used to

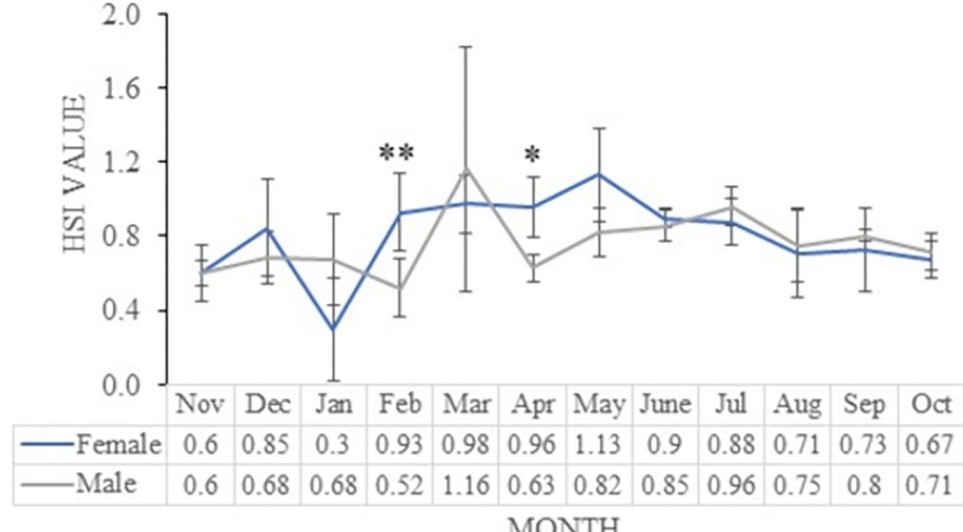

**Fig 3. Hepato-somatic index for male and female *M. aor* collected from Kaptai Lake, Bangladesh; data are presented as mean ± standard deviation (SD) (n = 5–9 for each sex in each month).** Asterisks denotes the significant difference between male and female (*, p < 0.05; and **, p < 0.01).

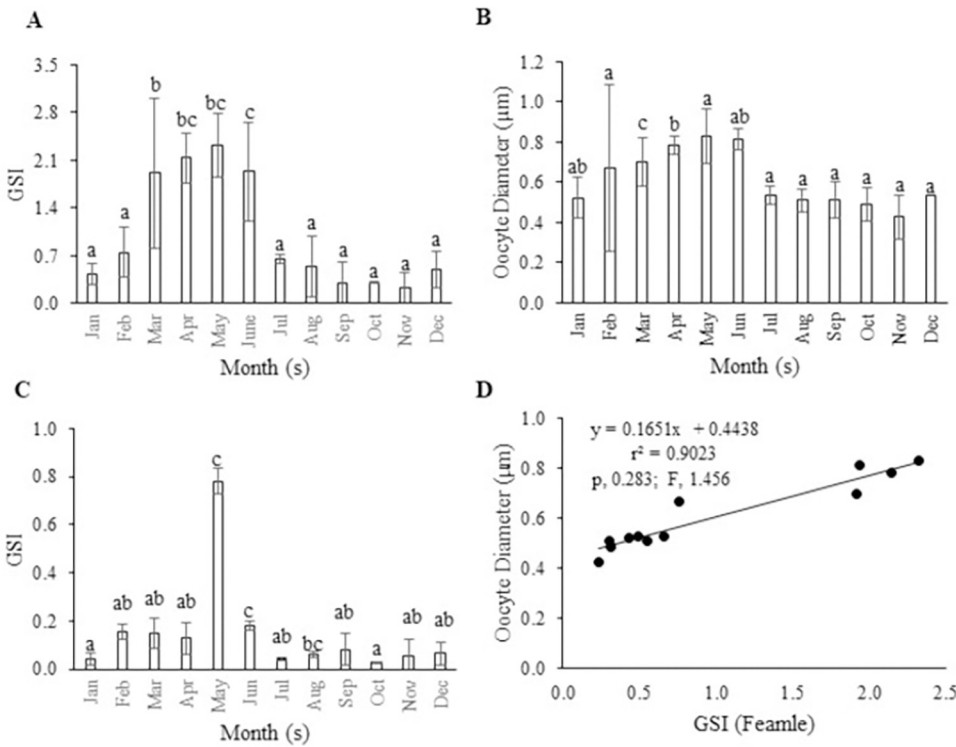

**Fig 4.** Gonadosomatic index (GSI) for (A) female and (B) male of *M. aor*; (C) oocyte diameter and (D) correlation between oocyte diameter and GSI of female *M. aor*, where the X-axis indicates the GSI values, and the Y-axis indicates the oocyte diameter (μm). Different subscripts of alphabet are statistically significant at p <0.05.

calculate the fecundity. The fecundity was found to vary from 18474 to 55504 eggs per female, with an average fecundity of 46989 eggs per female.

## Length at first sexual maturity (L$_m$)

*The L$_m$* for *M. aor* was calculated using the multiple relationships between TL vs GSI, TL vs MGSI, and TL vs DI (Fig 5). The GSI and MGSI in females were low ($< 2\%$) when fishes were

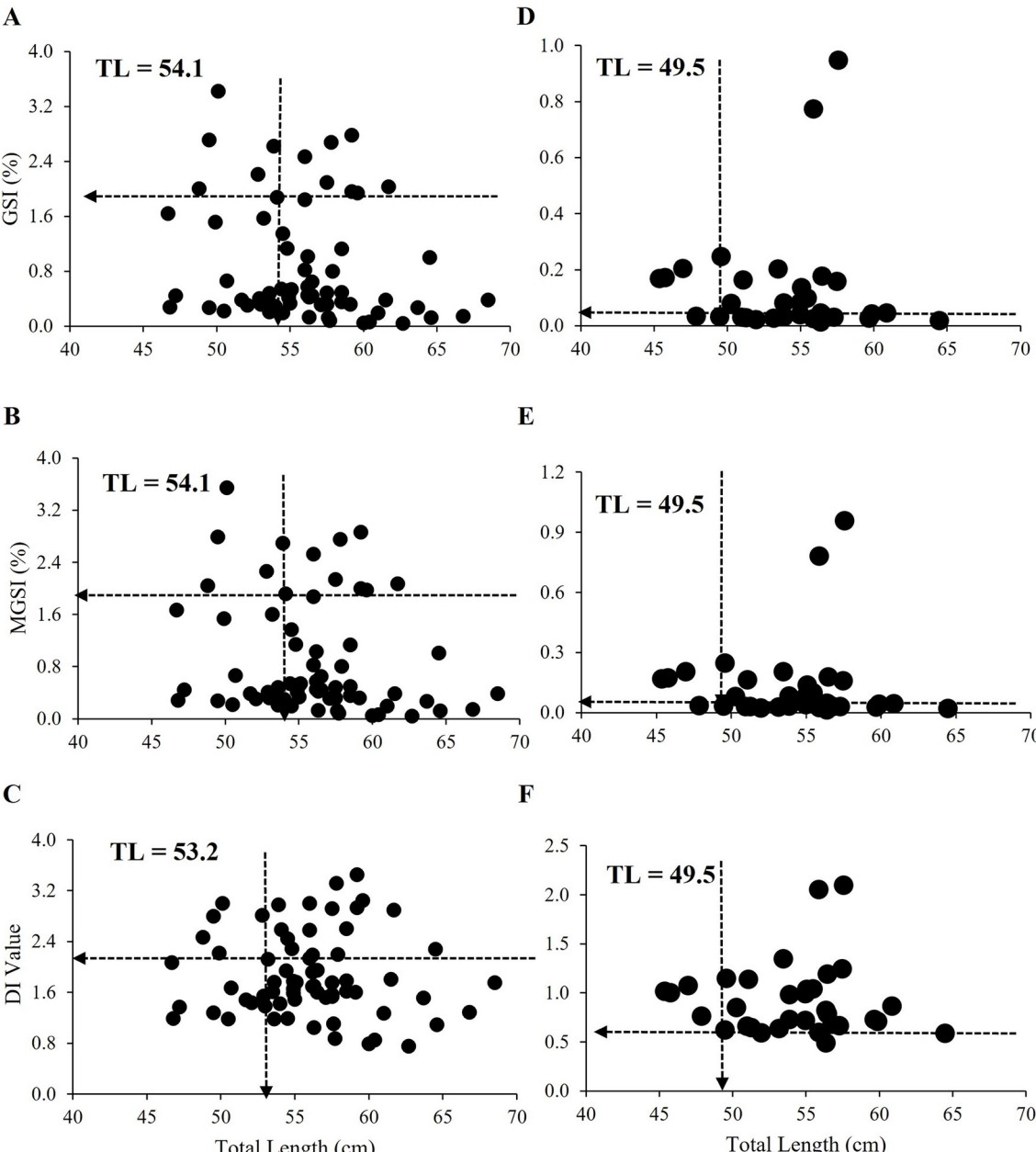

**Fig 5.** Relationship between (A) gonadosomatic index, GSI (%) and total length; (B) modified gonadosomatic index, MGSI (%) and total length; (C) Dobriyal index, DI and total length of female *M. aor* (n = 69) and (D) gonadosomatic index, GSI (%) and total length; (E) modified gonadosomatic index, MGSI (%) and total length; (F) Dobriyal index, DI and total length of male *M. aor* collected from Kaptai Lake, Bangladesh (n = 33).

less than 54.1 cm in TL. The DI was also low ($< 2$) at the same length. The relationship value between TL with GSI, MGSI ($> 2\%$) and DI ($> 2$) rose sharply around 54.1 cm TL in females, and hence that value was considered the $L_m$ for females (Fig 5A).

In the case of males, the GSI and MGSI were very low ($< 0.1\%$) for fishes smaller than 49.5 cm in TL (Fig 8). The DI value was also low ($< 0.1$) at the same length. The GSI, MGSI ($> 0.1\%$) and DI ($> 0.1$) rose sharply around 49.5 cm TL in males (Fig 5B).

## Gonadal maturation stages in females

**Pre-vitellogenic stage.** This stage was characterised by several oogonia (Oo), chromatin nucleolus (CN), early perinucleolus (EPO) and several primary growth oocytes. Oocytes were very small in diameter (0.51 μm ± 0.05), and the cytoplasm was densely stained with hematoxylin, as evidenced by a large and bright nucleus containing peripheral nucleoli. This stage was predominantly observed in August and September (Fig 6A and 6B).

**Early primary vitellogenic stage.** The ovary was larger with yellowish colouration. Early perinuclear oocytes (EPO) and late perinuclear stage oocytes (LPO) were abundant in this stage. Vascularisation in the ovary started to appear. This stage was found in October and November (Fig 6C and 6D).

**Advanced primary vitellogenic stage.** Zona radiata became visible in this stage. Yolk granules started to deposit. The number and size of yolk vesicles increased. Ova became visible to the naked eye, and vascularisation was prominent. This stage was seen in December and January (Fig 6E and 6F).

**Secondary vitellogenic stage.** The secondary vitellogenic stage was accompanied by the accumulation of eosinophilic yolk globules in the inner cortex. The cytoplasm was mostly covered with yolk globules. The nucleus contained some peripheral nucleoli. The zona radiata had increased its thickness. Ova in this stage were spherical and yellowish or orange in colour. Blood vessels appeared on the surface of the ovary. The number of ova could be counted by the unaided eye. This stage was found in February and March (Fig 7A and 7B).

**Ripe stage.** The ovary was largest, dense and light yellow, having GSI at its highest (2.31± 0.36). Vascularisation was highly conspicuous. The ovary occupied most of the body cavity. Oocyte diameter increased to the highest in this stage (0.83 μm ± 0.04). Post-ovulatory follicles were also present. This stage was seen in April and May (Fig 7C and 7D).

**Spent/regressing stage.** A sudden decrease in the ovary was seen. The ovary became a shrunken, hollow, sac-like structure. There were an abundant number of partially spent ovaries present. Oocytes in different vitellogenic stages were also found. This stage was found in June and July (Fig 7E and 7F).

## Gonadal maturity stages in males

**Immature.** Testes in this stage were small and creamy whitish. The seminal lobule of the testes contained many spermatogonia with few numbers of spermatocytes. Spermatogonia were spherical and stained with hematoxylin. Prevalence of primary spermatogonia was seen. The lumen of the tubules was imperceptible. This stage was found in September and October, with an average GSI value of 0.05 ± 0.03 (Fig 8A).

**Developing.** Testes were slightly larger, flat and translucent in colour. The germinal epithelium was seen throughout the testes in this stage. Spermatocyte formation started. Primary spermatocytes, secondary spermatogonia, and secondary spermatocytes were also seen. This stage was found in November and December with aGSI value of 0.06 ± 0.05 (Fig 8B).

**Pre-spawning.** Testes were in the late developing stages with spermatozoa in the lumens of the sperm ducts. All stages of spermatogenesis, like spermatogonia, spermatocytes and

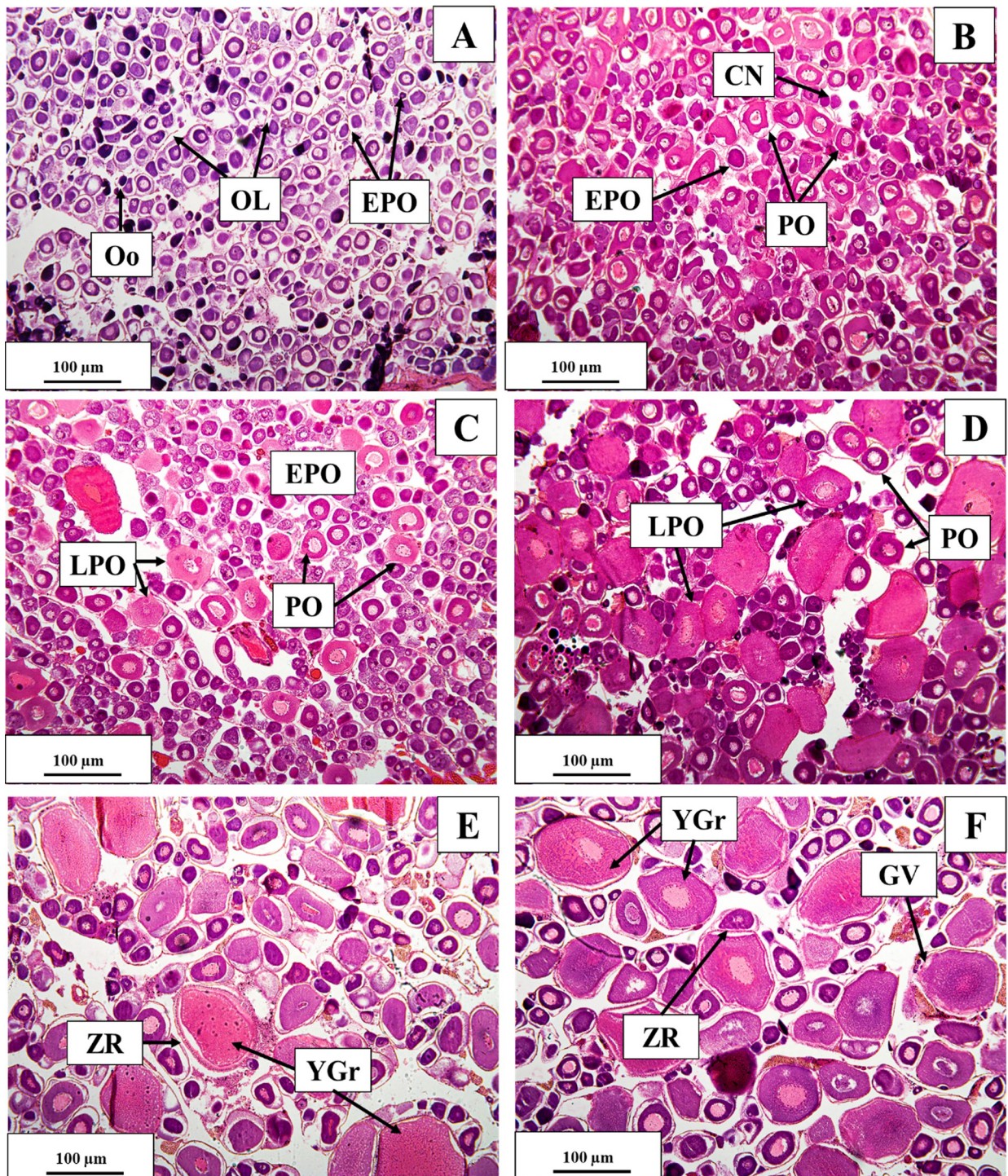

**Fig 6.** Ovarian maturation stages of *M. aor* collected from Kaptai Lake, Bangladesh, showing the previtellogenic stage (A, B), primary vitellogenic stage (C, D) and advanced primary vitellogenic stage (E, F); Oo = oogonia, OL = ovarian lamellae, CN = chromatin nucleolus, EPO = early perinuclear oocyte, PO = perinuclear oocyte, LPO = late perinuclear oocyte, YGr = yolk granule, GV = germinal vesicle, ZR = zona radiata; scale bar = 100 μm.

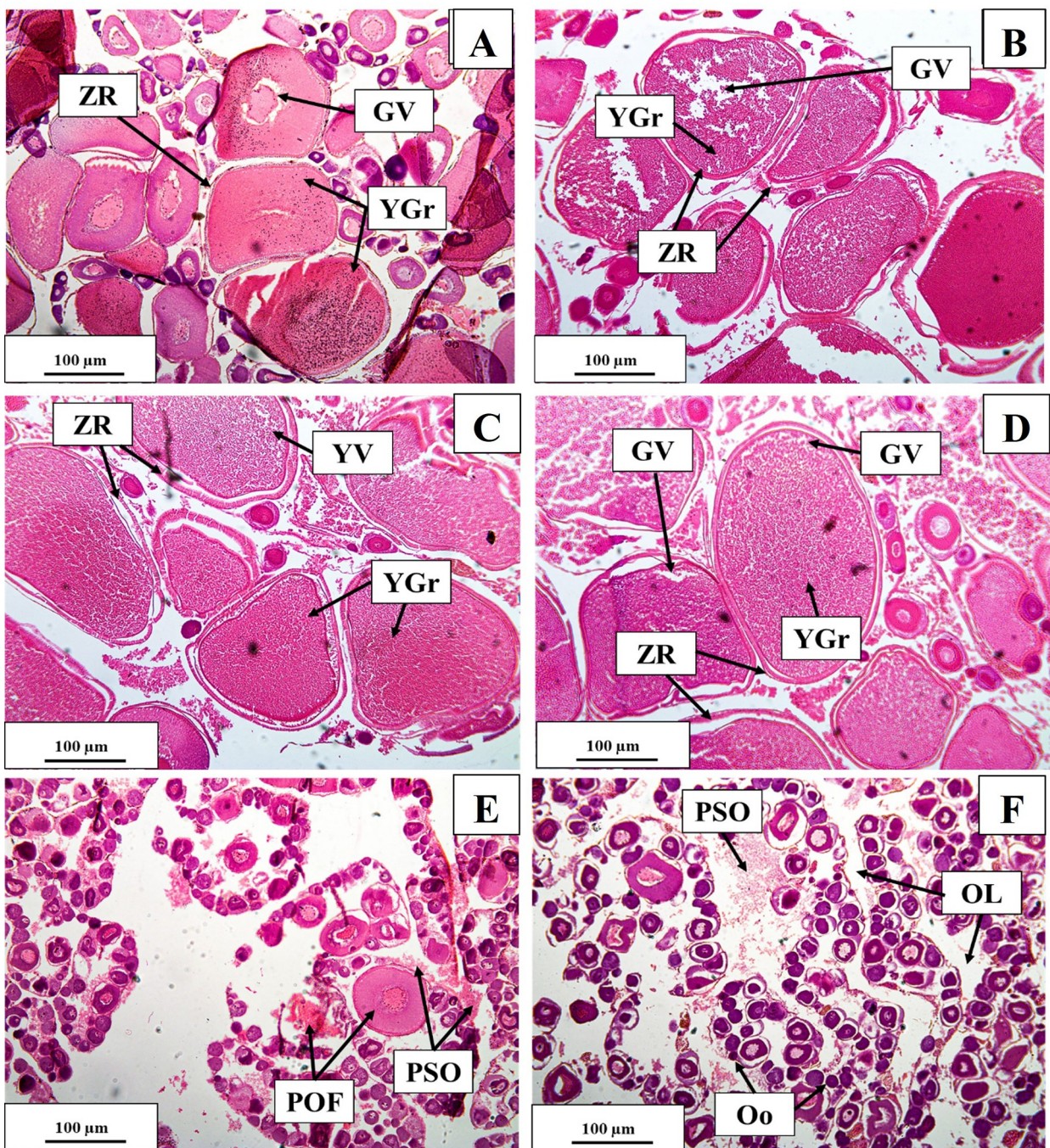

**Fig 7.** Ovarian maturation stages of *M. aor* collected from Kaptai Lake, Bangladesh, showing the secondary vitellogenic stage (A, B); ripe stage (C, D) and regressing stage (E, F); YGr = yolk granule, GV = germinal vesicle, YV = yolk vesicle, ZR = zona radiata. POF = post-ovulatory follicle, PSO = partially spent ovary, Oo = oogonia, OL = ovarian lamellae; scale bar = 100 μm.

spermatids, were usually seen in this stage, and among them, spermatocytes were predominant in the sperm tissue. Many cysts containing spermatocytes were seen throughout the seminal lobule. Testes were then creamy white and soft. In this stage, the GSI was 0.10 ± 0.06 and was predominantly seen in January and February (Fig 8C).

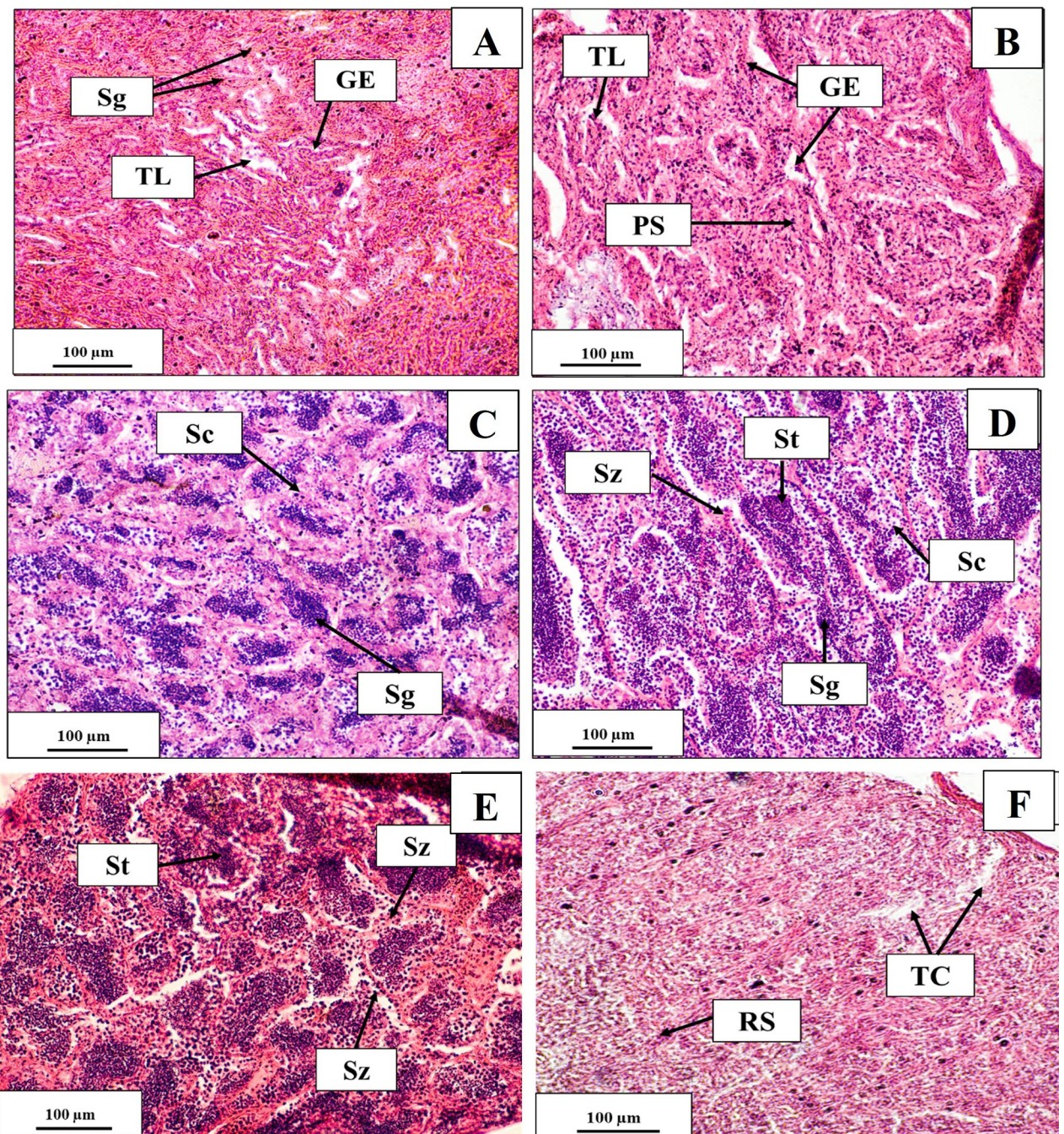

**Fig 8.** Testicular maturation stages of *M. aor* collected from Kaptai Lake, Bangladesh, showing the immature (A); developing (B); pre-spawning (C); ripe (D); spawning(E) and post-spawning (F) stages; TL = testicular lumen, GE = germinal epithelium, Sc = spermatocyte, Sg = secondary spermatogonia, St = spermatocyte, Sz = spermatozoa, RS = residual spermatid, TC = testicular cavity; Scale bar = 100 μm.

**Ripe.**   The number of spermatids in the seminal lobules and spermatozoa in the lumen was increased. The number of spermatogonia and spermatocytes decreased with the increase of the spermatids. The spermatozoa were spherical. This stage was found in March and April, with an average GSI of 0.14 ± 0.06 (Fig 8D).

**Spawning.**   Testes in this stage were large, maximum in GSI (0.48 ± 0.03), creamy white and soft. Histological slides were abundant with mature spermatozoa in the peripheral and central ducts of the testes and throughout their lumen. This stage was found in May and June (Fig 8E).

**Post-spawning.**   The GSI value decreased suddenly. Testes were found to be very thin and transparent. Some spermatozoa and residual spermatids were present, but empty spaces showed the regressing stage of the testes. This stage was found in July and August (Fig 8F).

## Variation in the environmental data (precipitation and temperature)

In the present study, monthly variation in precipitation level and water temperature was analysed and observed that the highest level of precipitation was found in June-August which coincides with the spawning periodicity of our studied species (Fig 9A and 9B). A strong correlation was observed between spawning and environmental parameters (Fig 9C and 9D). GSI value sharply increases with the beginning of rain and remained high during the full rainy season. Moreover, higher GSI also coincides with the increase in water temperature and found very low when temperature was minimum in the Kaptai lake. In addition, when compared with the two-time frame (year 1981–1990 and 2012–2021), precipitation levels significantly increased over 40 years in the last ten years from April to October (Fig 9A). In some cases, it increased 2 to 3-fold. In the case of temperature, a general trend of decreasing temperature was observed in the Kaptai lake ecosystem from the 1980s to 2012s. However, significant differences were only found in May ($p < 0.05$), June ($p < 0.01$) and November ($p < 0.05$) (Fig 9B).

## Discussion

Studies on reproductive biology are essential for sustainable conservation and management; also, they are a fundamental tool to assess the culture potential of a fish species, as they are assessments of the general circumstances of fish regarding reproductive biology [37–39]. This study observed the reproductive biology and gonadal maturation stages, along with some morphometric parameters such as the length-weight relationship, condition factor and relative condition factor, hepato- and gonadosomatic indices, length at first maturity ($L_m$), oocyte diameter and fecundity of *M. aor* from the Kaptai lake of Bangladesh in relation to the climatic events. The results show the breeding season to be in May and June, with good health and reproductive potentiality and better environmental conditions.

Length-weight relationships are employed to determine the weight associated with a given length. As direct weight measurements take time in the field, biomass is frequently calculated using length-weight regression coefficient (b) values. Positive allometric growth occurs when the value of 'b' is larger than 3.0, while isometric growth occurs when it is equal to 3.0. Conversely, negative allometric growth occurs when the value of 'b' is less than 3.0, and results in a population where fish become less rotund as length grows. The computed 'b' values for the male and female participants in the current study were 2.45 and 3.30, respectively, whereas it was 2.65 for the pooled data. For the pooled data and females, the 'b' value suggests a negatively allometric growth pattern, but for males, the growth is positively allometric. A lower exponential value in the female suggests a slower rate of growth in *M. aor*, whereas a higher exponential value in the male indicates a faster rate of weight gain relative to its length. This change in the b value was brought on by differences in the physiology of the male and female, the environment, the availability of food, and the ecological state of the habitats [31]. However, the combined data for *M. aor* revealed a negative allometric growth trend, indicating that weight gain in either species is not proportionate to growth in body length.

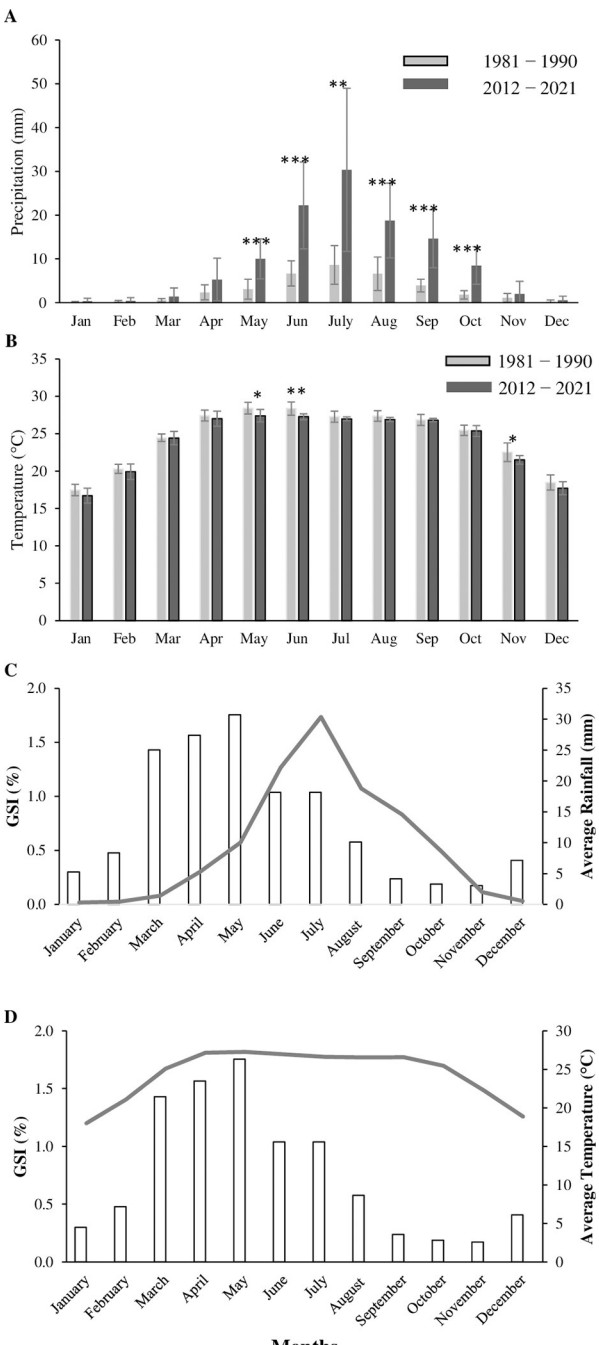

**Fig 9.** Time series data for (A) precipitation (mm) and (B) temperature (°C) from Kaptai Lake; values are presented as mean ± Standard deviation of the mean (SD). Asterisks denotes the significant difference between male and female (*, p < 0.05; **, p < 0.01 and ***, p < 0.001). Figure C and D indicates the relationship between GSI and rainfall respectively. Bar indicates the average value of GSI and line indicates the rainfall and/or temperature.

Growth in *M. aor* was revealed to be negatively allometric in males (b = 2.94), females (b = 2.94), and pooled data (b = 2.89) from Kaptai Lake in a previous study [26]. Additionally, a negative allometric growth pattern in *Sperata aor*, a species closely related to *M. aor* was reported [40]. However, Sani et. al., [41] observed an isometric growth pattern for *S. aor* in their research in the Betwa (2.98) and Gomti (3.02) rivers of India. The variations in sample

size, age or size group of the sample, geographic differences, and seasonal change may all contribute to the differences in 'b' values.

The condition factor is a measurement of the physiological status and general well-being of fish, and it considers things like maturity, spawning, environmental conditions, and availability of foods [42]. Information on seasonal variation in fish condition with respect to both the internal and exterior environment becomes crucial in fisheries biological research because the condition of fish can change substantially due to physiological, environmental, nutritional, and biological cycles. Since most fish do not comply with the cube law in their natural habitat, correlation and interpretation of these results can often be challenging. The function of fatness and gonad state are indicated by the condition factor, which can be better addressed by using the relative condition factor "Kn" [26]. The Kn value depends upon the stage and maturity of the gonads and the length of the fish [43]. A Kn value of more than 1 indicates good health of the fish, and a Kn value of less than 1 indicates poor condition of the fish [26].

In the present study, the K values of *M. aor* from Kaptai Lake highest in July and lowest in November, whereas the Kn value was highest in July and lowest in the month of December, which indicates more growth in the monsoon season and a less growth during winter. This is due to the suitable temperature, habitat condition and nutrition profile available during the monsoon season in comparison to the condition in winter in Kaptai Lake. The values of K and Kn in the present study indicate the good health condition of the fish in Kaptai Lake, as they are above 1. This finding coincides with the values of Kn of *M. aor* from Kaptai Lake [26] and the values of Kn for *S. aor* from the Gomti River [44].

The fecundity of a species is an essential aspect to consider when managing fish stocks. Fecundity is also a crucial biological trait for assessing the reproductive potential of fish [45]. By using randomly selected gravid females, fecundity was calculated from *M. aor*, and the fecundity of the fish was found to vary from 18474–55504 eggs per female, which weighed from 536.8–1480.4g. The fecundity of *M. aor* from the Kaptai reservoir in Bangladesh was recorded as 12560–48635 eggs per female [26]. Ranganathan and Radha, [46] reported the fecundity range varied from 21490 to 38400 eggs/female from the Bhabanisagar reservoir, India [30]. But Saigal [29] found the fecundity to be 45000–122500 eggs per female. In a recent study from North-East Bangladesh, the fecundity of *M. aor* varied from 59255 to70586, with an average value of 64920 eggs per female [47]. Variations in the fecundity might be due to the differences in stocks, sizes and nutritional status of fish and habitat characteristics [48].

The condition of the liver affects the metabolic activity and overall health status of fish. The ratio between liver weight and body weight is a sign of energy reserves in fish under varied environmental conditions. Therefore, the hepato-somatic index (HSI) is crucial for figuring out the physiological status of fish. In this study, the increased HSI values of males and females during the breeding months indicate a good reserve of lipid for reproductive expenditure. The HSI value rises when the food is readily available in large quantities and the environment is in suitable conditions. *M. aor* may have higher HSI values in May and March because of the good water quality, the surrounding environment and the availability of food. This is the first record of HSI of *M. aor* from the Kaptai Lake of Bangladesh, which could further help in sustainable management.

It is necessary to adjust the catch size group by regulating the mesh size and allowing the fish to reproduce for better stock maintenance [49]. Understanding the size of the fish at maturity and the season in which they reproduce is crucial for efficient fish population management and conservation [50]. The lengths at first maturity were 49.5 and 54.1cm in males and females, respectively. The $L_m$ of female *M. aor* was 48 cm According to Azadi et al. [25]. The $L_m$ for *S. aor* was 84 cm according to Saigal [29], and 57.3 cm according to Ramakrishniah [27], which is close to our findings.

The gonadosomatic index (GSI) is a good indicator of reproductive activity and is also related to the stages of gonadal maturation [31, 51]. It is used to improve the accuracy in determining the maturity stages and breeding time of fish species. In this study, a gradual increase in GSI was observed from December to February, reaching a peak in March to May and then decreasing from June to November. The GSI value was highest in May for both male and female *M. aor*, indicating the peak spawning season. Similar cyclic changes were found in *M. aor* from Kaptai Lake by Azadi et al. [26], with a prolonged peak time from April to July. In addition, a gradual increase in oocyte diameter was also observed in *M. aor* in association with the GSI as they approached gonadal maturity. The oocyte diameter and GSI values in *M. aor* delineate a homogenous growth pattern while reaching the highest oocyte diameter value in May ($0.830 \pm 0.109$ µm). Findings from earlier studies on *M. aor* from Kaptai Lake were comparable to our findings on oocyte diameter. However, the value was relatively higher in the study conducted by Saigal [29], where they found oocyte diameter ranged between 1.60 and 2.15 µm during spawning months.

The spawning season is essential for deciding when fish reproduce. The most accurate way to assess gonadal maturation is thought to be by histological examination [52]. The gonads of *M. aor* were examined histologically in the present study, and the stages were categorised following previous works [53, 54]. It was discovered that females had much more yolk granules from April to June. In April and May, mature eggs began to come out from the belly upon light pressure, which is a sign that the ovaries will begin to mature in the coming months. Multiple post-ovulatory follicles were noticed throughout the ovaries in June and July. The ovary included many atresia and empty areas according to histological studies obtained in July. In September, a small number of oogonia also started to grow, indicating the start of a new gonadal maturation cycle. The months of October and November are particularly rich with chromatin nuclear and perinuclear oocytes, which continued to grow until the beginning of February when yolk granules first appeared.

Male spermatogonia were prominent in September and October, and from November through February, they progressively began to mature into secondary spermatogonia and spermatocytes. In April and May, mature spermatozoa were detected in the peripheral and central ducts of the testes, as well as throughout their lumen, which suggests that males were spawning in this particular period. Numerous empty spaces with some spermatozoa still present in the testis were found in June and July, which is a sign that spawning was regressing. This finding predicts that gonadal maturation will peak between April and June, and then begin to decline.

In a previous study, Azadi et al. [26] reported that April to July was the breeding season by studying the oocyte diameter for *M. aor* in the Kaptai reservoir, Bangladesh, which coincides with our findings. A histological study on the gonadal development of *S. aor* from northeast Bangladesh revealed that the breeding season of *S. aor* lasts for a long duration from April to August and peaks in July to August in hatcheries [47]. Ramakrishniah [27] found a prolonged breeding season, starting from April to October. From the analysis of the environmental data, it is more obvious that there was a sharp difference in the precipitation rate (mm) over the past 40 years in the Kaptai Lake ecosystem. The average precipitation rates were significantly increased in recent decades during the month of May to September which is the spawning and juvenile nourishment period. Access precipitation may bring a lot of siltation to the Kaptai lake ecosystem from the upstream. As our studies fish is a bottom dweller, heavy siltation could damage the nesting and eggs which could be detrimental for their reproductive performance. In addition, higher precipitation may also cause a heavy water flow, which could wash out the eggs and larvae from Kaptai Lake, and that might be a major cause of the lower production rate of this species from this lake in recent years. The higher precipitation rate may be a crucial factor in regulating the reproductive seasonality and the major cause of differences in

the findings with the Azadi et al, (1992) from the same habitat [26]. However, there were no significant differences in the temperatures over the years (except in the May and June which may due to increase in the amount of rain) adjacent to the Kaptai Lake ecosystem (Fig 9B). According to Heggenes (1996), water velocity is considered the most significant environmental factor defining the habitat of stream-dwelling fishes [55]. A previous study on salmonids showed that salmonid eggs require sufficient flow to ensure their oxygen demands and may also be harmed or swept out of beds during strong flow events, which will cause them to die [56]. It was also found that environmental enrichment influences the ovarian development of *Sperata* sp. with significantly increasing the weight gain rate, GSI and HSI [57].

The advent of the rains and the changes in hydrology are critical in the reproductive cycle of fishes. However, their relationship with the reproductive biology, commencement of spawning and migration is still poorly understood. Previous studies on carp have shown that temperature and rainfall directly alter fish spawning behaviour [58]. In another study, de Magalhães Lopes et al., [59] demonstrate that changes in rainfall, hydrological parameters and lunar phage determine the spawning migration in neotropical fish. The correlation between rainfall and commercial catch (mullet and barramundi in Australia) implies that fisheries may be vulnerable to the effects of climate change [60]. In the present study, we found a three-fold increase in rainfall which could also result from climate change and may alter the spawning and overall fishery of *M. aor* in the Kaptai lake. To sustain the fisheries production, necessary steps should be taken in the Kaptai lake ecosystem. Currently, overexploitation is one of the major threats in Kaptai Lake in addition to climate change. Massive numbers of fries and fingerlings of large fish are being caught with destructive fishing gear. To be specific, mosquito seine net (Kechki jal) damages ichthyo-planktons, especially IMC's fries, resulting in decreased production of IMCs and other large fishes like *M. aor* [61]. Moreover, anthropogenic and human-made pollutants (during navigation, marketing, tourism) are also a crucial threat to the Kaptai lake. The principal sources of pollution at Kaptai Lake include waste dumping, open defecation by slum inhabitants, locals, and industries, the use of fertilizers and pesticides around the lake, and the rampant discharge of oily chemicals by engine boats and passengers on water transport. The practice of catch limits, habitat restoration, public awareness, restricting tourism and protecting breeding grounds along with a fishing ban from April to June could be the key strategies for sustainable management of this species in Kaptai Lake.

## Conclusion

The knowledge obtained through histological studies of gonads of *M. aor* provides the information that the breeding period of this species is from April to June, with a peak in May in Kaptai Lake, which will further help in the sustainable management of this species. In addition, the overall abiotic factors like pH, DO, alkalinity, free $CO_2$ etc. in the Kaptai lake [61] and the health status of this commercially important species is in a good state to be reproductively active in the lake ecosystem which further indicates the strong correlation between rainfall and spawning in this spawning. In fisheries research, stock assessment, and management, reproductive biology is crucial. Sustainable, productive fisheries and aquaculture enhance income and improve livelihoods, promote economic growth, and safeguard the environment and natural resources, all while improving food and nutrition security. A sustainable approach to fisheries and aquaculture will aid in the conservation of natural resources and the preservation of fish populations for future generations.

Because the reproductive features of any species influence its intrinsic capability and sustainability of exploitation, open water fisheries management heavily relies on information about distinct stages of reproductive development of any fish population. Fishing laws must be

based on a thorough understanding of the reproductive cycles and population dynamics of the area's commercially significant species. Furthermore, a detailed comprehensive study on the effect of rainfall and water flow should be evaluated to better understand the reproductive success, embryo survival, and offspring fitness for this species, as well as other commercially important species from Kaptai Lake.

## Acknowledgments

The authors also express sincere thanks to the fishers and fish sellers who helped to collect samples and all of the laboratory personnel of the Fish Biology and Biotechnology Department, CVASU, for their kind support.

## Author Contributions

**Conceptualization:** Shifat Ara Noor, Fatema Akhter, Md. Moudud Islam.

**Data curation:** Mst. Halima Khatun.

**Formal analysis:** Shifat Ara Noor, Md. Mahiuddin Zahangir.

**Funding acquisition:** Md. Moudud Islam.

**Investigation:** Shifat Ara Noor, Muhammad, Md. Moudud Islam.

**Methodology:** Shifat Ara Noor, Fatema Akhter.

**Project administration:** Md. Main Uddin Mamun.

**Resources:** Mst. Halima Khatun, Md. Mahiuddin Zahangir, Md. Moudud Islam.

**Software:** Md. Mahiuddin Zahangir.

**Supervision:** Md. Moudud Islam.

**Writing – original draft:** Shifat Ara Noor.

**Writing – review & editing:** Shifat Ara Noor, Muhammad, Md. Main Uddin Mamun, Fatema Akhter, Mst. Halima Khatun, Md. Mahiuddin Zahangir, Md. Moudud Islam.

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
