## [Decision Letter · Decision Letter 0]

11 Sep 2024

PONE-D-24-35998Reproductive biology of long-whiskered catfish (Mystus aor) relative to temperature and precipitation from the Kaptai Lake of BangladeshPLOS ONE

Dear Dr. Islam,

Thank you for submitting your manuscript to PLOS ONE. After careful consideration, we feel that it has merit but does not fully meet PLOS ONE’s publication criteria as it currently stands. Therefore, we invite you to submit a revised version of the manuscript that addresses the points raised during the review process.

We look forward to receiving your revised manuscript.

Kind regards,

Ishtiyaq Ahmad, Ph.D

Academic Editor

PLOS ONE

Journal Requirements:

“CVASU-University Grant Commission”

“The authors wish to express their gratitude to the UGC-CVASU of Bangladesh FY 2020– 21 for funding this research. The authors also express sincere thanks to the fishers and fish sellers who helped to collect samples and all of the laboratory personnel of the Fish Biology and Biotechnology Department, CVASU, for their kind support.”

“CVASU-University Grant Commission”

5. Please provide a complete Data Availability Statement in the submission form, ensuring you include all necessary access information or a reason for why you are unable to make your data freely accessible. If your research concerns only data provided within your submission, please write "All data are in the manuscript and/or supporting information files" as your Data Availability Statement.

Reviewers' comments:

Reviewer's Responses to Questions

**Comments to the Author**

1. Is the manuscript technically sound, and do the data support the conclusions?

Reviewer #1: Yes

2. Has the statistical analysis been performed appropriately and rigorously? 

Reviewer #1: Yes

3. Have the authors made all data underlying the findings in their manuscript fully available?

Reviewer #1: Yes

4. Is the manuscript presented in an intelligible fashion and written in standard English?

Reviewer #1: Yes

5. Review Comments to the Author

Reviewer #1: Detailed observations on Q. 1 to 4 above have been included in the attached document. However, here I am pasting them again.

This review concerns manuscript PONE-D-24-35998 titled "Reproductive biology of long-whiskered catfish (Mystus aor) relative to temperature and precipitation from the Kaptai Lake of Bangladesh." My comments follow.

Overall summary:

The study addresses an important topic in fisheries biology and is crucial for fisheries management, conservation, and aquaculture development, especially in regions where fish populations are under pressure from overexploitation and environmental changes.

A clear justification has been provided for the study by highlighting the economic importance of the study species Mystus aor, the declining fish stocks in Kaptai Lake, and the lack of recent studies on this species in the context of changing climatic conditions. The need for updated biological data to inform conservation and management strategies has been well articulated.

Established methods have been used for analyzing reproductive biology parameters namely length-weight relationships, condition factors, gonadosomatic index, hepatosomatic index, fecundity, and histological analysis of gonads.

Data have been collected over a full year cycle to capture seasonal variations.

Long-term environmental data (40 years) have been analyzed to contextualize the findings and appropriate statistical analyses like ANOVA, t-tests, and correlation analyses have been used.

Specific comments:

Sample size: Monthly about 136 fish samples were collected [Line 174]. It remains unclear how many fish were analyzed for each studied parameter. The study could benefit from a more detailed explanation of how the sample size was determined and put to use to capture seasonal and interannual variability. There is no mention of whether the sample size was sufficient to detect meaningful differences in the parameters measured. This information would help in assessing the robustness of the findings.

Abiotic parameters: While precipitation and temperature data are analyzed, other potentially important factors, for example, dissolved oxygen, pH, water flow rates, etc. Have not been considered.

Correlation between environmental factors (e.g., precipitation) and reproductive parameters (spawning) would benefit if presented in the form of a statistical analysis.

Given the multiple parameters measured, a multivariate analysis might have provided additional insights into how different factors interact to influence reproductive biology.

Confounding factors: While the findings mention overfishing and pollution as threats, very limited discussion is provided on how these factors might have influenced their results.

The discussion would benefit if a broader comparison with similar studies or species is included.

References:

1.Missing in-text citation: [22] Talwar and Jhingran [Line 623].

2.Missing in ‘References’: [27] Ranganathan and Radha [Line 438].

Spellings:

1.‘Bagridae’ NOT ‘Bagaridae’ [Line 143].

2.‘recorded’ NOT ‘recored’ [Line 438].

6. PLOS authors have the option to publish the peer review history of their article (what does this mean?). If published, this will include your full peer review and any attached files.

Reviewer #1: **Yes: **Ravindra A. Pawar

---

## [Author Response · Author response to Decision Letter 0]

18 Sep 2024

Respond to reviewers comment is attached in file section

---

## [Editor Report · Decision Letter 1]

15 Oct 2024

Reproductive biology of long-whiskered catfish (Mystus aor) relative to temperature and precipitation from the Kaptai Lake of Bangladesh

PONE-D-24-35998R1

Dear Dr. Islam,

We’re pleased to inform you that your manuscript has been judged scientifically suitable for publication and will be formally accepted for publication once it meets all outstanding technical requirements.

Kind regards,

Ishtiyaq Ahmad, Ph.D

Academic Editor

PLOS ONE

---

## [Editor Report · Acceptance letter]

30 Oct 2024

PONE-D-24-35998R1 

PLOS ONE

Dear Dr. Islam, 

I'm pleased to inform you that your manuscript has been deemed suitable for publication in PLOS ONE. Congratulations! Your manuscript is now being handed over to our production team.

Kind regards, 

on behalf of

Dr. Ishtiyaq Ahmad 

Academic Editor

PLOS ONE